# Functional Cheeses: Updates on Probiotic Preservation Methods

Hannah Caroline Santos Araujo, Mônica Silva de Jesus, Rafael Donizete Dutra Sandes, Maria Terezinha Santos Leite Neta and Narendra Narain *

Laboratory of Flavor and Chromatographic Analysis, Federal University of Sergipe, Av. Marcelo Deda Chagas, s/n, Jardim Rosa Elze, São Cristóvão 49100-000, SE, Brazil; hcarol197@gmail.com (H.C.S.A.); monicasj.sst@gmail.com (M.S.d.J.); rafael.donizete.dutra@gmail.com (R.D.D.S.); terezinhaleite@gmail.com (M.T.S.L.N.)
* Correspondence: narendra.narain@gmail.com; Tel.: +55-79-3194-6514

**Abstract:** The consumption of natural products, especially those that promote some health benefit, has become a choice for consumers. Foods that improve health when ingested are called functional foods. Among them, the most consumed are probiotics, which are defined as microorganisms that, when administered in adequate quantities, can promote a health benefit for consumers. Probiotic dairy products, especially cheese, are the most appreciated and have been produced to possess the properties that favor the viability of these microorganisms after passing through the gastrointestinal tract. They provide benefits such as antibacterial activity, prevention of cancer and cardiovascular diseases, anti-obesity effects, anti-diabetic effects, improved immune systems, and prodiseases, among others. Despite cheeses being a viable matrix for the survival of these probiotics, the development and adoption of technologies with the aim of increasing the viability of probiotic bacteria are necessary, which presents a research trend to be increasingly explored, as well as studies about the complex mechanisms of health benefits resulting from the actions of probiotics. Thus, this review aims to address the most recent innovations regarding the incorporation of probiotics in cheeses and their functional potential.

**Keywords:** cheese; probiotic; innovations; functional foods

## 1. Introduction

It is clear that consumer preference for a food product that promotes some final health benefits is growing more and more due to the strong relationship between diet and human health. Functional products are excellent options, as they aim to improve quality of life and combat nutrition-related diseases. Due to the numerous benefits related to this type of food, they are increasingly part of consumers' daily lives, promoting the growth of this niche market. The development and innovation of food products, that have nutritional properties and promote the improvement of consumers' health, are growing more and more in recent times. Functional foods, such as probiotics, are already known to consumers and scientifically recognized for their beneficial effects on health [1,2].

Probiotics have the basic definition of being microorganisms that, when administered in adequate quantities, can promote a health benefit for consumers. Among the benefits associated with its consumption are protection against many pathogens, lowering cholesterol, improving the immune system, and even improving lactose intolerance. Probiotic products, especially dairy products, are widely consumed worldwide; among these functional dairy products, cheese stands out in this market sector [3,4].

As one of the most consumed dairy products from a nutritional point of view, cheese is an excellent source of proteins, minerals, and vitamins, and hence has a high nutritional value. These nutrients present in the constitution of cheese favor the survival of probiotic microorganisms, as they protect them from the high acidity of the intestinal environment [5]. Due to the versatility of cheeses and their consumption across age groups, the development

of functional cheeses has been a successful topic addressed by several articles found in the literature [6–11].

Nonetheless, even with characteristics that favor the growth of probiotic microorganisms in cheese, the biggest challenge in probiotic research is to maintain its activity during the whole shelf life of a product, especially in cheese, due to a series of factors, such as the acidic environment, presence of oxygen, antimicrobial compounds, and ingestion in the gastrointestinal tract. In addition to these obstacles, due to the cheese matrix, some strains of probiotics are susceptible to producing off-flavors when multiplying, consequently resulting in reduced sensory acceptance by the consumer of the probiotic product. Therefore, it is necessary to develop technologies that guarantee the therapeutic properties of probiotics throughout their useful lives.

Currently, studies focused on probiotic cheeses are highlighting their use to promote a health benefit for individuals suffering from diseases, acting on some harm caused by them [5,6,11]; the incorporation of ingredients that act to benefit the functionality of this dairy product [10,12]; the use of technology to develop protective agents against these microorganisms in the gastrointestinal tract [13]. Therefore, this review is aimed at promoting the use of cheeses as vehicles for probiotics, considering their effects as functional foods, in addition to presenting an overview of the last 5 years regarding the progress of research involving techniques to improve the viability of microorganisms and their beneficial effects on the health of the host.

## 2. Functional Foods

It is common knowledge and agreement that food is a necessity and essential to life which meets the nutritional needs of each human being. Food intake and an individual's health status are directly interconnected and their eating habits can directly affect their metabolism. Nutrients such as fats, carbohydrates, and proteins provide energy for the growth and maintenance of the body [14,15]. Consumers are increasingly aware of their food choices and looking for a higher-quality eating habit. Due to this growing demand for nutrient-rich foods, there are foods that provide us with other benefits in addition to nutrition; these are functional foods [15–17].

Functional foods do not have a standard definition. According to the Food and Agriculture Organization (FAO) of the United Nations, functional foods are defined as foods that contain, in addition to nutrients, other components that may be beneficial to health [18]. The pioneering country to propose regulatory legislation for functional foods was Japan; later, other countries also developed their own regulations for this type of food, such as the US Federal Food and Drug Administration in the United States (US), the Food Safety and Standards Authority of India (FSSAI) in India, the China Food and Drug Administration (CFDA) in China, the European Union (EU) in Europe, and the National Health Surveillance Agency (NSSA) in Brazil [14].

In addition to having added nutritional value, functional foods help to promote optimal health conditions and can reduce the risks of one or more non-communicable diseases, such as cancer, type 2 diabetes, and cardiovascular diseases. For a food to be considered functional, it must be validated in laboratory tests to comply with the regulations determined by each country [19]. The production of these foods led the food industry to expand research and development activities [19,20]. As a highly targeted sector, the global functional foods market was valued at US$178 billion in 2019, reaching around US$258.8 and US$281.4 billion in 2020 and 2021, respectively. By 2028, a growth of approximately 10% is estimated, with revenues of US$530 billion [21,22].

This intense innovation in the area of functional foods paved the way for the study of the addition of components such as bioactive compounds, probiotics, prebiotics, phytochemicals or herbs, natural antioxidants, and bioactive peptides, which in turn give the product a functional characteristic [14]. Although certain foods are rich in nutrients that benefit health, there is a growing interest in developing processed functional products. The

most common functional foods found in the markets are yogurts, cereals, cheese, bars, and protein drinks.

Among these products, functional foods, those containing probiotics, represent a large industrial sector. The growing interest in probiotics is directly linked to the rapid and profitable expansion of the development of foods containing these microorganisms. These food components offer a series of health benefits and positively affect the intestinal microbiota [19,23].

## 3. Probiotics

The human gastrointestinal tract is the habitat of several bacteria, where a symbiotic relationship is maintained with the rest of the body. The intestinal microbiota is a complex ecosystem composed of microorganisms linked to various nutritional, metabolic, endocrine, psychological, and immunological mechanisms. These microorganisms have the functions of maintaining the intestinal barrier, nutrient metabolism, synthesis of bioactive compounds and vitamins, fermentation of non-digestible carbohydrates, and mainly immunomodulation. The so-called "barrier effect" that prevents pathogens from affecting the gastrointestinal tract can be strengthened by taking dietary supplements such as probiotics [24,25].

In recent decades, the global consumption of probiotics has increased, arousing the interest of companies. Industrial interest in these probiotic bacteria is directly linked to a rapid, profitable expansion in the functional foods and supplements sector. The global probiotics market has an estimated growth rate of 3.72% in the period between 2021 and 2026. The probiotic yogurt market reached a profit of US$ 33,248 million in 2021. Other probiotic dairy products also stood out, with a growth of US$7418 million in 2021, with a projected growth of 3% between 2021 and 2026 [26,27].

As previously mentioned, probiotic dairy products have a large representation within functional products. The highlight of these products is their multiple benefits as vehicles for probiotics and advances in their technology. Dairy products are an extremely beneficial matrix for these microorganisms, as their fermentation process facilitates the optimization of the viability of these bacteria. The storage conditions of these products also facilitate the survival of these microorganisms [26]. The beneficial effects of probiotics in these foods depend on the number of viable cells present in them after the gastrointestinal tract; therefore, to be considered a probiotic food, it must contain concentrations between $10^8$–$10^9$ UFC/mL [28]. Moreover, the viability of the product is a determining factor in achieving health effects [29].

*Probiotic Strains*

Some microorganisms are quite well known to have probiotic action. Some of them are described in Table 1. Most traditional strains are *Lactobacillus* and *Bifidobacterium* genera, which are present in the flora of gut microbiota. For a strain to be considered probiotic, it has to have some characteristics like resistance to gastric acid and bile acid, good adherence to human mucus and/or human epithelial cells, and antimicrobial activity against pathogenic bacteria or fungi [30,31].

**Table 1.** Common industrial strains of microorganisms used as probiotic starter cultures.

| Genus | Species | pH | Temperature Range (°C) | Applications | Reference |
|-------|---------|-----|------------------------|--------------|-----------|
| *Lactococcus* | *Lactococcus lactis* ssp. *lactis* | 4.6 | Tmi: 8; Ti:30; Tma: 40 | Mesophilic probiotic starter used for different types of dairy products such as cheese (Gouda and Edam). | [30,32] |
| | *Lactococcus lactis* ssp. *lactis* biovar *diacetylactis* | 4.6 | Tmi:8; Ti:22–28; Tma:40 | | [30,32] |
| | *Lactococcus lactis* ssp. *cremoris* | 5.6 | Tmi:4; Ti:20–28; Tma:37 | | [30,32] |
| *Streptococcus* | *Streptococcus thermophilus* | 4.5 | Tmi:22; Ti:40; Tma:52 | Thermophilic probiotic starter used mainly for yogurts | [30,32] |
| *Lactobacillus* | *Lactobacillus acidophilus* | 4.2 | Tmi:27; Ti:37; Tma:48 | Mesophilic probiotic starter used for different types of dairy products, especially in cheeses during ripening | [30,32] |
| | *Lactobacillus delbrueckii* ssp. *bulgaricus* | 3.8 | Tmi:22; Ti:45; Tma:52 | | [30,32] |
| | *Lactobacillus delbrueckii* ssp. *lactis* | 3.8 | Tmi:18; Ti:40; Tma:50 | | [30,32] |

**Table 1.** *Cont.*

| Genus | Species | pH | Temperature Range (°C) | Applications | Reference |
|---|---|---|---|---|---|
| | *Lactobacillus helveticus* | 3.8 | Tmi:22; Ti:42; Tma:54 | - | [30,32] |
| | *Lacticaseibacillus casei* | - | Ti:30 | - | [30,32] |
| | *Lactiplantibacillus plantarum* | - | Tmi:12; Ti:37; Tma:40 | - | [33] |
| | *Lacticaseibacillus rhamnosus* | - | Ti:37 | - | [34] |
| *Leuconostoc* | *Leuconostoc mesenteroides* ssp. *cremoris* | 4.5 | Ti:35 | Mesophilic starter culture used for cheeses like sour cream | [30] |

Tmi: minimum temperature; Tma: Maximum temperature; Ti: ideal temperature.

The most famous probiotic microorganisms are of the genus *Lactobacillus*, which are a major part of the LAB. *Lactobacillus* can convert hexose sugars to lactic acid, which is the main component needed to produce dairy products and is responsible for producing the lactase enzyme that breaks down lactose or milk sugar. It also produces lactic acid, which helps control the proliferation of harmful bacteria and works as a muscle fuel, helping the body absorb different minerals. However, *Lactobacillus* can be used for other applications, such as the production of sauerkraut, pickles, sourdough, wine, and other fermented products. In all these products, in addition to converting hexoses to lactic acid, *Lactobacillus* can create a hostile environment for spoilage microorganisms, which in turn improves the preservation of food. *Lactobacillus* flora is commonly found in the human gastrointestinal tract and the female genitourinary tract. These microorganisms help the organism protect itself from chronic diseases since they can produce enzymes that display antibiotic, anticancer, and immunosuppressant properties [35].

*Lactobacillus* species can be classified into three main groups, which are obligatory homofermentative, facultative heterofermentative, and obligatory heterofermentative. This classification is defined based on sugar metabolism and the final product generated from fermentation. Species such as *Lactobacillus delbrueckii* subsp. *bulgaricus*, *Lactobacillus lactis*, *Lactobacillus helveticus* (thermophilic), *Lacticaseibacillus rhamnosus*, and *Latiloactobacillus curvatus* (mesophilic) belong to the group of homofermentative *Lactobacillus* that use only hexoses as a carbon source to produce lactic acid as their only or main product. Among the functions of these species, their important role in the cheese maturation process can be highlighted, being frequently present in their secondary microbiota [36].

In the group of facultative heterofermentative *Lactobacillus*, the main products generated are organic acids, carbon dioxide, alcohol, and other carbon sources and hexoses. Within this group are bacteria such as *Lacticaseibacillus casei*, *Lacticaseibacillus paracasei*, and *Lactiplantibacillus plantarum*, which in turn are not found in starter cultures, but are associated with secondary fermentations, which have a beneficial effect during the cheese curing process. These microorganisms are generally found in artisanal starter cultures and are therefore called NSLAB—non-starter acid lactic bacteria [36].

Another known probiotic microorganism is the *Bifidobacterium* genus which includes more than 90 species, excluding unclassified species. This microorganism was first isolated from breast-fed infant feces [37]. Six species of *Bifidobacterium*: *Bifidobacterium lactis*, *B. breve*, *B. bifidum*, *B. infantis*, *B. adolescentis*, and *B. longurn* are very important in probiotic dairy production. Just like the genus *Lactobacillus*, Bifidobacterium has probiotic activity due to its beneficial effects such as anti-infection and anti-depression, regulating the host immune system, and facilitating host nutrition adsorption in both humans and animals.

Many studies demonstrate that *Bifidobacterium* has an effective anti-inflammatory action like *B. longum* ATCC 15708 that showed antimicrobial activity against many pathogens, including *Escherichia coli* O157:H7 ATTC 35150, *Salmonella typhimurium* ATTC 13311, and *Listeria monocytogenes* ATTC 19115. The antimicrobial activity of B. longum BB536 protects against gut-derived sepsis caused by *Pseudomonas aeruginosa*, likely by interfering with the adherence of pathogens to intestinal epithelial cells [37,38]. As an example, other studies have shown the anticancer activity of *Bifidobacterium*. The work detailed by Chen et al. [37] presented that dietary supplementation of *B. longum* BB-536, which significantly inhibits the 2-amino-3-methylimidazo quinoline (IQ)-induced incidence of the colon (100% inhibition),

liver (80% inhibition) tumors in male rats, suppresses the induced mammary carcinogenesis (50% inhibition), and liver carcinogenesis (27% inhibition) of female rats.

## 4. Probiotic Cheese

Being one of the most consumed dairy products worldwide, the cheese production process and consumption dates back to 5.200 BC, being present in people's eating habits for centuries. Cheese means a product obtained from whole milk, standardized or skimmed, coagulated by enzymes, or by acidification and heating. Thus, cheese is a mass of casein and milk fat, among other components that are present in milk, with the addition of yeasts, salt, dyes, and other substances allowed under current legislation. The cheese-making process is based on three stages, namely, acidification, coagulation, and dehydration, which is the removal of whey [39,40].

Cheese is a food rich in nutrients and can be consumed in the form of a dessert, a snack, in the natural form, or even being incorporated as an ingredient in a product. Cheese is a dairy product that has a very diverse microbial flora derived from its raw material. Lactic acid bacteria (LAB), which may be present in milk or added during cheese processing, are responsible for assigning different sensory characteristics to each type of cheese. This is due to the complex interaction of these microorganisms with the proteins, carbohydrates, and fats of milk [36,41].

Because cheese is a good source of protein, vitamins, and minerals, the main ones being calcium and phosphorus, cheese consumption has been growing due to the healthy and positive image that is linked to incorporating this food into the diet [42]. In 2017, 156.9 million tons of milk were produced in the European Union, where 98.9% of this production was converted into dairy products, with 37% destined for cheese production. In European countries where cheese consumption was higher in 2019, approximately 1938.60 thousand tons were produced in France, 2297.40 thousand tons in Germany, and 1327.30 thousand tons in Italy [43]. Regarding consumption in the EU, around 10,425.66 thousand tons were consumed, followed by the United States with 6331.67 thousand tons consumed in 2019 [44].

Due to the constituents of dairy products being sources of essential nutrients, such as proteins and calcium, they become important vehicles for the addition of probiotics, thus increasing their functional capacity. These foods are the main providers of probiotic bacteria for the gastrointestinal tract, having the advantage of favoring the survival of these microorganisms during the digestive process [45,46]. Among dairy products, cheese is the most efficient carrier of probiotic microorganisms, as it has a buffer function that is effective against the acidic conditions of the digestive process. The selection of probiotic strains is carried out based on the type and manufacturing process of each cheese [3,11].

Currently, in the literature, it is possible to find several publications that report the addition of different strains of probiotic microorganisms to different types of cheese, thus giving the food a functional characteristic. These cultures are added together with the starter culture when the milk reaches the appropriate temperature for processing according to the type of cheese [1,5,9,11,47]. In Table 2, it is possible to check the different types of probiotic cheeses studied, together with the types and quantities of probiotic cultures added.

**Table 2.** Probiotic microorganisms used in cheese production.

| Type of Cheese | Probiotic Microorganism Used | Type of Probiotic Culture | Quantity | References |
|---|---|---|---|---|
| Minas artisanal cheese | *Lactiplantibacillus plantarum* and *Lacticaseibacillus rhamnosus* | Isolated culture | $10^8$ cfu/g | [48] |
| Prato cheese | *Lacticaseibacillus casei*-01 | DVS Culture | $10^7$–$10^8$ CFU/g | [6] |
| Processed cheese ("requeijão cremoso") | *Bacillus coagulans* MTCC 5856, *Bacillus coagulans* GBI-30 6086, *Bacillus subtilis* PXN 21, *Bacillus subtilis* PB6 and *Bacillus flexus* HK1 | Isolated culture | $10^6$–$10^7$ spores/g | [49] |

**Table 2.** *Cont.*

| Type of Cheese | Probiotic Microorganism Used | Type of Probiotic Culture | Quantity | References |
|---|---|---|---|---|
| Feta | *Lactobacillus acidophilus* and *Bifidobacterium animalis* | DVS Culture | $10^{10}$ CFU/mL | [31] |
| Mozzarella cheese | *Lactobacillus acidophilus* | Isolated culture | $10^{10}$ CFU/g | [50] |
| Ricotta Cheese prepared from buffalo milk | *Lactobacillus acidophilus* La-05 | DVS Culture | $10^{8}$ CFU/mL | [47] |
| Cheddar prepared from buffalo milk | *Lactobacillus acidophilus* and *Bifidobacterium bifidum* | DVS Culture | $10^{8}$–$10^{10}$ CFU/g | [1] |
| Minas Frescal Cheese | *Lactococcus lactis* NCDO 2118 | DVS Culture | $10^{7}$–$10^{8}$ CFU/g | [5] |
| Spreadable ricotta cheese | *Lacticaseibacillus paracasei* BGP1 | DVS Culture | $10^{10}$ CFU/mL | [8] |
| Dutch Edam cheese | *Lacticaseibacillus casei* LAFTI-L26 | DVS Culture | $10^{8}$ CFU/mL | [51] |
| Wagashi cheese | *Lacticaseibacillus rhamnosus* and *Lactiplantibacillus plantarum* | DVS Culture | $10^{8}$–$10^{9}$ CFU/mL | [9] |
| Processed cheese | *Lactiplantibacillus plantarum* NRC AM10 and *Limosilactobacillus reuteri* NBIMCC 1587 | Isolated culture | 3% (1:1) | [52] |
| Cream cheese | *Lactiplantibacillus plantarum* CCMA 0359 | Isolated culture | $10^{10}$ CFU/g | [13] |
| Kariesh cheese | *Bifidobacterium lactis* BB-12, *Lacticaseibacillus rhamnosus* NRRL B-442 and *Lactobacillus gasseri* NRRL B-14168 | Isolated culture | 3% | [53] |
| Petit Suisse | *Lactobacillus acidophilus* LA5 and *Bifidobacterium animalis* subsp. *lactis* BB12 | DVS Culture | $10^{10}$ CFU/g | [54] |
| Fresh cheese prepared from skimmed milk | *Lactiplantibacillus plantarum* CNPC 003 | Isolated culture | $10^{9}$ CFU/L | [55] |

CFU—Colony-forming Unit.

Sameer et al. [47] studied the incorporation of *Lactobacillus acidophillus* La 05 in the matrix of ricotta made from buffalo milk and its influence in terms of compositional, texture, color, and sensory parameters. It was verified that the incorporation of this microorganism does not alter these parameters. The cell count of this microorganism was $7.8 \pm 0.2$ log CFU/g by product. Scanning electron microscopy revealed that the ricotta matrix consists of a network of proteins, a network that provides the microorganism with additional protection. Hurtado-Romero et al. [54] evaluated the use of blueberry by-products to develop an ingredient with nutritional and technological potential, increasing the functional value of probiotic Petit-suisse cheese, containing the strains of *Lactobacillus acidophilus* LA5 and *Bifidobacterium animalis* ssp. *lactis* BB12, and concluded that blueberry pomace powder presented a high value of bioactive compounds, thus increasing the potential of this cheese and providing the consumer with a product beneficial to health.

Amiri et al. [51] investigated the effects of the adjunct culture of *Lacticaseibacillus casei* LAFTI-L26, brine concentration (0, 3, and 5%), brining time (1, 2, and 3 days), milk treatment, and ripening time (1, 31, and 61 days) on the physicochemical, microbial, textural, microstructural, and sensorial properties of traditional Edam cheese. During the days studied, this probiotic strain presented a viability of 9.95 log CFU/g. Furthermore, the addition of the probiotic strain *Lacticaseibacillus casei* LAFTI-L26 improved the proteolysis, lipolysis, and sensory characteristics of Dutch Edam cheese.

## 5. Technology for Preserving Probiotics in Cheese

The biggest challenge in probiotic research is to maintain its activity during the whole shelf life of the product, especially in cheese, due to a series of factors, such as the acidic environment, the presence of oxygen and antimicrobial compounds, and ingestion in the gastrointestinal tract. In addition to the susceptibility of some strains of probiotics to produce off-flavors when multiplying, this implies a reduction in sensory acceptance by the consumer of the probiotic product. Therefore, it is necessary to develop technologies to ensure the therapeutic properties of probiotics throughout their shelf life.

One of the techniques that can be used to preserve probiotics in food products is microencapsulation which can protect probiotics against the adverse conditions to which they can be exposed during the manufacture and storage of the food products and, finally, their targeted release in the lower part of the gastrointestinal tract [46]. Microencapsulation has been considered an efficient and novel technique for improving the viability of probiotics in both food products and the intestinal tract, as survival is essential for probiotic microorganisms targeted to populate the host gut to provide health benefits [56].

The microencapsulation of probiotics is a process that retains them in a polymeric membrane, protecting them, and in some cases, allowing programmed release in the gastrointestinal tract under specific conditions. This process can be affected by various encapsulation parameters, such as the concentration of wall material, carrier size, cell load, encapsulation time, and the viability of probiotic cells in both the food and gastrointestinal tract [46]. Different techniques can be used to microencapsulate probiotics, such as extrusion, spray drying, spray coating, emulsion, coacervation, and immobilization in fat and starch. The choice of the technique that can be used will depend on the probiotic microorganism and the final product. Some of the techniques that can be used are presented in the next paragraphs. A diagram of the application of these techniques can be seen in Figure 1.

### 5.1. Extrusion

Extrusion is a physical method that is based on mixing probiotic microorganisms with a hydrocolloid solution to form a suspension, which is then extruded through a nozzle at high pressure using a syringe or appropriate droplet-generating devices. The droplets produced are placed in a gelling bath, and from this direct contact, the cells become trapped in a three-dimensional grid that occurs from the ionic cross-linking of the polymer, thus forming gel spheres that can be incorporated into food matrices [46,57].

Alginates and other natural polysaccharides are often used as hydrocolloid materials because they do not cause damage to probiotic cells, thus resulting in high viability in the food product and prolonged survival of encapsulated cells under storage and simulated gastric and enteric environments. As for the gelling (or hardening) solution, it consists of divalent cautions, such as magnesium or, more commonly, calcium [46,57,58].

### 5.2. Spray-Drying

Spray drying is one of the most commonly used encapsulation techniques for food ingredients. The encapsulation of probiotics using this method is based on the emulsion or dispersion of the strains in an aqueous solution with the encapsulating agent, followed by spraying the mixture through a nozzle and evaporating the water from the contact of the droplets with air or hot gas, which results in a dry particulate powder with low water activity [46,57].

Studies on the encapsulation of microorganisms and probiotic products by spray drying have increased, and different authors have suggested that this process has improved the stability and viability of probiotics and is more efficient in the long-term preservation of lactic acid and other probiotic cultures encapsulated with different support materials [46]. However, the effectiveness of encapsulation depends on different factors, such as probiotic characteristics (probiotic strain and its growth phase); dryer characteristics, and spray drying conditions (flow rate, chamber humidity, chamber inlet and outlet temperatures);

the addition of encapsulating material and/or protectants (trehalose, non-fat milk solids (MSNF), adonitol, acacia, granular starch, prebiotics, soluble fiber, maltodextrin, gum Arabic, etc.) and solution viscosity, and so on [46,57,59].

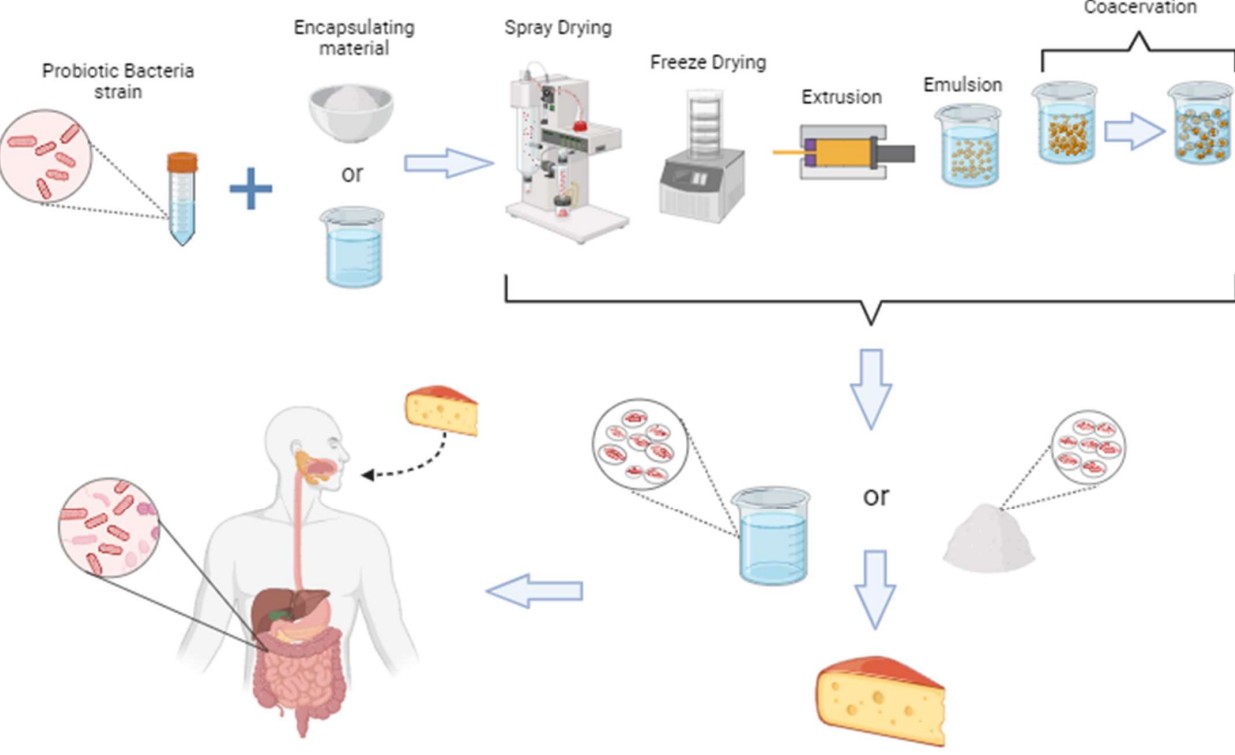

**Figure 1.** Schematic summary of microencapsulation techniques.

### 5.3. Lyophilization/Freeze Drying

Freeze drying is one of the most commonly used techniques for dehydrating probiotic cultures and other thermally sensitive components, and is based on the freezing and subsequent sublimation of the frozen ice present in the molecules to the gaseous state under vacuum and low temperatures [57,60]. Freeze-dried probiotic microorganisms are very stable, have a high survival rate (higher compared to other methods, such as spray drying), and have been extensively used in many dairy products [46].

Despite the effectiveness of the method, freeze-drying affects the state of cell membrane lipids and the structure of sensitive cellular proteins, thus having a negative effect on the survival of the probiotic [57]. Studies are being carried out to improve the efficiency of lyophilization of these strains, and one of the alternatives is the combination of lyophilization with suitable encapsulation agents and/or cryoprotectants [57,58]. One of these examples is the first work carried out that reported the encapsulation of *Lactobacillus* F19 and *Bifidobacterium* Bb12 in enzymatically gelled sodium caseinate microcapsules, followed by freeze-drying [61].

### 5.4. Emulsion

Emulsification is a technique that can be used in food processing to encapsulate probiotic microorganisms. The principle of this technique is the dispersion of a liquid (a small volume of a water-based polymeric suspension containing probiotic cells) in another immiscible liquid (a large amount of oil/organic phase) followed by mechanical homogenization in the presence of an emulsifier to form a water-in-oil emulsion. Gel spheres are formed in the oil phase as the water-soluble biopolymer becomes insolubilized [57].

The emulsion system methods that can be used to encapsulate probiotics can be divided into three categories: (a) emulsification and ionic gelation; (b) emulsification and

enzymatic gelation; and (c) emulsification and interfacial polymerization. Ding et al. [62] generated W/O/W emulsions with the internal aqueous phase loaded with probiotic cells and various protective compounds. The oil phase consisted of soybean oil and polyglycerol polyricinoleate (PGPR), a common food-safe molecular surfactant, stabilizing the W1/O and O/W2 interfaces [62,63].

*5.5. Coacervation*

Coacervation is a highly promising and widely used encapsulation technology for probiotics and consists of three steps: the formation of immiscible phases, the deposition of the coating, and the solidification of the coating. Probiotic cells are added to a hydrocolloid, and the main polymers used as encapsulating agents are proteins and various polysaccharides [57].

This technology can be conducted by two methods: simple and complex coacervation. The first consists of evaporating the solvent that surrounds the molecules of a colloid through the addition of another non-electrolytic solvent in which the colloid is insoluble. Complex coacervation consists of the combination of two hydrocolloid solutions with opposite charges, causing the interaction and precipitation of complex polymers [46,56]. Some parameters must be considered for the method to be effective in encapsulation, such as the total percentage of solids, the dispersion conditions (ionic strength, pH, temperature, etc.), and the characteristics of the encapsulating materials and core ingredients (weight molecular, conformation, charge density, and composition) [57].

Each of the encapsulation techniques mentioned presents different principles and different advantages and disadvantages, which can be seen in Table 3 for preserving probiotic strains.

**Table 3.** Disadvantages and advantages of encapsulation techniques.

| Technique Encapsulation | Disadvantages | Benefits | Reference |
|---|---|---|---|
| Extrusion | • Formation of large particles (2–5 mm); <br>• Difficult to apply on a large scale due to the slow formation of particles; <br>• Lower survivability during storage and lower encapsulation efficiency. | • Simplicity and easy handling; <br>• Low cost on a small scale; <br>• Soft operating conditions. | [46,57] |
| Emulsion | • Residues left on the surface of the particles; <br>• Large variation in particle size and shape. | • Applicable at the industrial level; <br>• Formation of particles with a diameter of less than 300 μm; <br>• High yield. | [46,57] |
| Coacervation | • Expensive and complex method; <br>• It is necessary to dry the coacervated probiotics; <br>• It is necessary to use different dehydration methods. | • Soft preparation conditions; <br>• High encapsulation efficiency; <br>• Allows the incorporation of a substantial number of probiotic cells. | [57] |
| Spray drying | • Requires strict control of drying conditions; <br>• High temperatures can result in losses due to dehydration thermal and osmotic stress; <br>• Possibility of early release of probiotics due to the damage that can be caused to the surface structure of cells. | • The most economical method; <br>• Short processing time; <br>• High efficiency; comprises a single unitary process; <br>• High performance and easy application; <br>• Easy storage; provides small capsules (diameters less than 100 μm); <br>• Can be combined with other methods. | [46,57,59,64,65] |
| Freeze-drying | • Long processing time; <br>• High cost; <br>• Damage to the cell wall may occur due to the formation of ice crystals during freezing. | • High stability; <br>• Higher survival rate compared to spray drying. | [46,57] |

Table 4 presents some publications in the last 5 years focused on the use of encapsulation techniques to protect probiotics in cheeses.

**Table 4.** Application of preservation technology in probiotic cheeses.

| Type of Cheese | Probiotic Strain | Preservation Technology | Encapsulating Material | Process Conditions | Application Stage | Storage Time (Days) | Main Results | Log CFU/g/log CFU/mL | Reference |
|---|---|---|---|---|---|---|---|---|---|
| Cream cheese | *Lactiplantibacillus plantarum* | Spray-drying | Whey powder | T = 150 °C | After homogenization of the mass | 90 | High viability at the simulated gastrointestinal tract; It did not alter the organoleptic properties of the cheese. | I = 8.69; F = 8.31 | [13] |
| Chami (traditional soft cheese) | *Pediococcus pentosaceus* | Freeze-drying | Camel milk proteins and wheat starch | T = −80 °C | After homogenization of the mass | 9 | Camel milk proteins revealed higher cell viability (98.6%) than probiotic cells encapsulated with wheat starch (70.7%). | I = 9.23/9.57; F = 9.10; 6.77 | [66] |
| Kariesh cheese | *Bifidobacterium lactis* BB-12, *Lacticaseibacillus rhamnosus* NRRL B-442 and *Lactobacillus gasseri* NRRL B-14168 | Extrusion | Sodium alginate and rice flour | Rice flour (1.3 and 5%); 0.2 M of CaCl$_2$ solution | After homogenization of the mass | 21 | The survival rate of probiotics when exposed to *in-vitro* simulated gastrointestinal solutions was recorded at 72.9, containing 5% of rice flour. | I = 8.59, 8.69, 8.87; F = 5.39, 6.33, 6.89 | [53] |
| White soft cheese | *Bifidobacterium lactis* BB12 | Extrusion | Sodium alginate, fish oil, and pomegranate peel extract (PPE) | Fish oil (5 and 10%); PPE (36.55 µg/mL); 0.2 M of CaCl$_2$ solution | Before coagulation | 30 | The probiotic + fish oil + PPE emulsion protected the probiotic bacteria during storage for 30 days. | I = 8.08; F = 5.5 | [67] |
| Feta cheese | *Lactobacillus acidophilus* and *Bifidobacterium animalis* | Emulsification method | Sodium alginate and pectin | Solutions of sodium alginate or pectin (2% *w/v*) | After pressing and cutting the mass | 45 | Sodium alginate proved to be more efficient in relation to the viability of probiotics compared to pectin. | I = 9.3; F = 8.6 | [31] |
| Cream cheese | *Enterococcus durans* | Freeze-drying | Maltodextrin and cryoprotectant (sucrose and lactulose) | T = −50 °C | After homogenization of the mass | 92 | *E. durans* strains show high resistance to the freeze-drying process in the presence of all cryoprotectants used, and the best results were obtained when lactulose, a prebiotic sugar was used. | I = 8.38; F = 8.04 | [68] |
| Goat Ricotta | *Lactobacillus acidophilus* (La-05) | Ionic gelation | Alginate and chitosan | Solution of CaCl$_2$ at 0.5 mol/L | After salting the mass | 7 | Microencapsulation of probiotic cultures resulted in increased probiotic survival. | I = 7.18; F = 6.88 | [7] |
| Reino cheese | *Lactobacillus acidophilus* (LA-3) | Ionic gelation | Biopolymeric solution (ascorbic acid (0.04% *w/v*), L-cysteine hydrochloride (0.04% *w/v*), and sodium alginate (4% (*w/v*)) | Solution of CaCl$_2$ at 4% (*w/v*) | After heating the milk | 25 | High viability in the cheese containing the microcapsules | I = 9.34; F = 8.49 | [69] |

**Table 4.** *Cont.*

| Type of Cheese | Probiotic Strain | Preservation Technology | Encapsulating Material | Process Conditions | Application Stage | Storage Time (Days) | Main Results | Log CFU/g/log CFU/mL | Reference |
|---|---|---|---|---|---|---|---|---|---|
| Processed cheese (Requeijão cremoso) | *Lactobacillus acidophilus* | Spray chilling | | Air pressures (2.5 kgf/cm$^2$) | Added at the beginning of the mass heating process | 90 | The formulation containing microcapsules showed greater sensory acceptance of texture and probiotic counts greater than 6 log CFU/g during storage and simulation of gastrointestinal conditions. | I = 7.52; F = 7.15 | [70] |
| Iranian white cheese | *Lactiplantibacillus plantarum* ATCC 8014 | Liophilization and spray drying | Whey protein isolate (WPI) and Gum Arabic (GA) | T = −80 and 100 °C | After heating the milk | 61 | High survivability of *L. plantarum* ATCC 8014 in freeze-dried microcapsules than in spray-dried microcapsules during storage time (60 days). | I = 9.11; F = 6.44 | [71] |

T: Temperature; I: Initial quantity of cells; F: Final quantity of cells; CFU—Colony-forming Unit.

Mudgil et al. [66] evaluated the effects of microencapsulation of probiotic microorganisms on the stability and survival capacity of Chami (Cheese) as a carrier matrix. Through microencapsulation by freeze-drying, with the use of camel milk proteins and wheat starch as encapsulating material, they demonstrated that this technique for maintaining the viability of the probiotic microorganism *Pediococcus pentosaceus*, in Chami cheese, proved to be efficient, where the microorganism presented a value of 109 log CFU/g after passage through the gastrointestinal tract. El Sayed et al. [53] evaluated the viability of microencapsulated probiotic strains of *Bifidobacterium lactis* BB-12, *Lacticaseibacillus rhamnosus* NRRL B-442, and *Lactobacillus gasseri* NRRL B-14168 under simulated gastrointestinal conditions within UF-Kariesh cheese. The microorganisms were microencapsulated using the extrusion technique, where sodium alginate and rice flour were used as encapsulating materials, and it was observed that the survival rate of probiotics when exposed to simulated gastrointestinal solutions in vitro varied between 59.23 and 72.91%, demonstrating that microencapsulation was efficient.

de Andrade et al. [13] studied the influence of microencapsulation, using the spray-drying technique, on the survival of the *Lactiplantibacillus plantarum* CCMA 0359 strain in cream cheese during storage of the cheese. They found that microencapsulation significantly increased cell protection after the simulated gastrointestinal tract, showing a cell count greater than 8.31 log CFU/g during storage of this cheese.

It is notable that the various encapsulation techniques demonstrated efficiency in preserving probiotic strains in cheeses. Simulated digestions (in vitro) reported in the literature demonstrate the efficiency of different encapsulation techniques in preserving probiotic strains in cheese; however, in vivo studies need to be carried out to truly confirm the effectiveness of these techniques on their survival.

## 6. Other Ways of Incorporating Probiotics into Cheese
*Edible Coatings*

The incorporation of probiotics in cheese can be achieved through the techniques already mentioned in this work; however, a technique used to give the cheese a functional characteristic is the use of edible probiotic coatings in this dairy product. The use and development of edible coatings is an alternative that meets consumer demand for healthier foods that contain health-promoting components. Regarding the application of edible coatings containing probiotics, studies are focused on the application of whole and minimally processed fruits [72,73]. However, only a few studies are found in the literature applying biofilms containing probiotics to cheese, with the aim of evaluating the viability of these microorganisms as well as the quality of this dairy product. Some of these works are described below.

Saez-Orviz et al. [74] developed a symbiotic coating, containing the probiotic strain *Lactiplantibacillus plantarum* CECT 9567, and a prebiotic compound LBA (lactobionic acid). This coating was applied to cottage cheese, where concentrations of this probiotic strain were monitored during storage and simulated in vitro digestion. Regarding the viability of this probiotic strain, the coating samples PRO (*Lactiplantibacillus plantarum* CECT 9567), SYN2 and SYN4 (symbiotics) reached final concentrations of 6.53, 6.72, and 7.23 log UFC $g^{-1}$, respectively, with the best result being obtained in the SYN4 sample (40 g $L^{-1}$ LBA + 109 CFU $mL^{-1}$ probiotic strain). In relation to the results obtained in simulated digestion, the bacteria inside the PRO, SYN2, and SYN4 coatings survived, with an average value of 8.53, 9.06, and 9.54 log UFC $g^{-1}$, respectively, showing that the coating exerted a protective effect.

Celay and Atasoy [75] determined the optimal concentration of fructooligosaccharide (FOS) inulin (IN) in edible films containing *Bifidobacterium animalis* subsp. *lactis* BB-12[®] based on sodium caseinate, investigating the use of these film formulations for coating sliced processed cheese. With respect to the viability of the probiotic strain, in the edible film samples containing 0% and 1% IN, the addition of 3% FOS resulted in a higher concentration of *B. animalis* subsp. *lactis* BB-12[®]. Furthermore, 2% and 3% FOS provided the greatest bacterial protection in samples containing 2% and 3% IN. Furthermore, 2% and

3% FOS provided the greatest bacterial protection in samples containing 2% and 3% IN. The viability of *B. animalis* subsp. *lactis* BB-12® was generally increased by the addition of IN, but the authors reported that this increase was not significant ($p < 0.05$), except for film samples containing 2% FOS.

Olivo et al. [76] evaluated the edible coating (film-forming solution) using a mixture of sodium alginate, calcium chloride, and water as a vehicle for probiotic bacteria in cheeses. The samples were then subjected to four treatments: uncoated cheeses (SEMc), cheeses coated with sodium alginate (AG), cheeses coated with sodium alginate and L. acidophilus (0.001%) (AGLA), and cheeses coated with sodium alginate and *L. helveticus* (0.001%) (AGLH) and these products were analyzed for 15 days. Evaluating the viability of probiotic microorganisms, they reported that the lactic acid bacteria (LAB) count values showed that there was initial microbiota in both coated and uncoated cheeses, with a greater value in treatments for cheese with sodium alginate and *L. acidophilus* (7.71 log CFU/g), with only a decrease of 1 log CFU/g on the fifteenth day.

Guadalupe et al. [77] characterized edible films based on potato starch and fermented and non-fermented whey solutions by *Lacticaseibacillus rhamnosus* and *Lactobacillus acidophilus* and applied these edible films to Manchego-type cheese, evaluating its quality during storage in two different packaging. With respect to the results, they reported that linear low-density polyethylene packaging together with edible films maintained the moisture content, weight, and color characteristics of the cheese during storage time. Additionally, the edible film significantly increased the probiotic content of the cheese without interference from packaging and gastrointestinal simulation, maintaining a viable count of both *L. acidophilus* and *L. rhamnosus* (>7 logarithmic cycles) after 14 days of storage.

In the works reported above, we can observe another approach to incorporating probiotic microorganisms into cheeses. Undoubtedly, the use of edible probiotic coatings in this dairy product is an efficient tool to offer the consumer a functional food not only capable of resisting the gastrointestinal tract but also delivering a stable, quality, and safe cheese for consumption.

## 7. Current Applications of Probiotics in Cheese as Health Promoters

Probiotics are microorganisms that promote beneficial effects on health. However, their functionality essentially depends on their ability, after being ingested orally, to survive passage through the gastrointestinal tract due to the acidic conditions of the stomach, bile salts, antimicrobial compounds, and other adverse conditions [71,78,79]. Furthermore, probiotics are expected to be able to multiply quickly, have the efficient capacity for adhesion, colonization, and persistence in the gastrointestinal tract to maintain their metabolic activity, compete adequately with resident microbiota or other pathogens for nutrient absorption, improve the immune system, and promote balance in the intestinal microbiota [79,80].

Many factors influence the viability of probiotic bacteria, including the composition of the food into which the probiotics are inoculated and the processing conditions [71]. Cheese is one of the best and most favorable food matrices for the incorporation of probiotics. In addition to being one of the most consumed dairy products in the world, it has a high buffering capacity for milk proteins, a high-fat content, lactose as a fermentable sugar, and the dense structure of this dairy matrix, which encourages the survival of probiotic strains during storage and through passage in the gastrointestinal tract [81–83].

Probiotic strains have specific functional properties, and their mechanisms of action may vary from one probiotic strain to another, but in most cases, a combination of activities is likely. Although it has been proven that probiotics have many therapeutic effects on consumers' health, several studies have been carried out in order to expand knowledge about the functional properties of probiotic cheeses [78,79,84].

Among the benefits already reported in the literature (Table 5), which can be mentioned are digestive regulation, stimulation of the immune system, reduction of cholesterol levels and oxidative stress, antibacterial activity, prevention of cancer and cardiovascular diseases, anti-obesity effect, antidiabetic effect, reduction of symptoms of lactose intolerance, and

allergic diseases, among others [12,63,71,85,86]. In current diseases such as COVID-19, the consumption of probiotics has also been shown to be efficient in regulating the balance of the intestinal microbiota [87].

**Table 5.** Health benefits of probiotic cheeses.

| Type of Cheese | Probiotics Strains | Objective | Beneficial Potential | Reference |
|---|---|---|---|---|
| Prato | *Lacticaseibacillus casei*-01 | Efficacy of repeated consumption of probiotic Prato cheese against the inflammatory and oxidative condition induced by cigarette smoke in a mouse model | Repeated intake of probiotic cheese reduced oxidative stress in the lungs, intestine, and liver, and alleviated inflammation in the lungs. | [6] |
| Kalari | *Lactobacillus plantarum* (NCDC 012), *Lacticaseibacillus casei* (NCDC 297), *Levilactobacillus brevis* (NCDC 021) | Evaluate the in vitro anti-proliferative, immunomodulatory, and antidiabetic potential of Kalari cheese incorporated with probiotics | The addition of probiotics enhanced the antiproliferative (against human breast and colon cancer cells, neuroblastoma), antidiabetic, antimicrobial, and immunomodulatory activity of the Kalari cheese | [88] |
| Minas Frescal (Brazil) | *Lactococcus lactis* NCDO 2118 | To investigate the probiotic therapeutic effects of a Minas Frescal cheese containing *L. lactis* NCDO 2118 on ulcerative colitis induced by dextran sodium sulfate in mice. | Mice that consumed the probiotic cheese exhibited reduced severity of colitis, with attenuated weight loss, lower disease activity index, limited shortening of the colon length, and reduced histopathological score. | [5] |
| Minas Frescal and Prato (Brazil) | *Lacticaseibacillus casei*-01 | To evaluate the effect of different probiotic dairy matrices on antihyperglycemic activity in vitro and in vivo. | Prato cheese showed greater anti-hyperglycemic activity in vitro (greater inhibitory activity of α-amylase and α-glucosidase) and in vivo [less increase in postprandial glycemia and maintenance of other glycemic indices in healthy individuals. | [89] |
| Chami | *Pediococcus pentosaceus* | Incorporation of microencapsulated probiotic strain in Chami, and evaluation of antidiabetic activity in vitro | Chami fortified with encapsulated probiotic bacteria exhibited greater retention of bioactive properties, in terms of inhibition of α-glucosidase and Dipeptidyl peptidase IV (DPP-IV) during storage. | [66] |
| Cheddar | *Lactobacillus helveticus* 1.0612, *Lacticaseibacillus rhamnosus* 1.0911, *Lacticaseibacillus casei* 1.0319 | To evaluate the influence of digestion and the addition of different probiotics in cheddar cheese regarding the degree of proteolysis and the inhibitory activity of the angiotensin-converting enzyme (ACE) | Cheddar cheese with different probiotics contributed to the release of ACE-I peptides and in vitro digestion further increased their activity in cheese samples. | [90] |
| Fresh cheese | *Lactiplantibacillus plantarum* 299v, *Bifidobacterium animalis* Bo | To evaluate the potential of probiotic cheese fortified with bioactive fatty acids using in vitro models with an emphasis on modulating obesity-related metabolism and the immunomodulatory response. | The combination of the strains with the fatty acids in the cheese provided an increase in bacterial survival during passage through the gastrointestinal tract, indicating a possible synergistic effect between both. The digested fractions also stimulated the production of adipokines, reduced lipid accumulation in hepatocytes, increased adipolysis, and had an anti-inflammatory effect. | [83] |
| Fresh cheese | *Lactococcus lactis* LB1022, *Lactiplantibacillus plantarum* LB1418 | To evaluate the effect of probiotic cheese on inducing alcohol metabolism | Intake of probiotic cheese improved alcohol metabolism, regulated fatty acid oxidation, and prevented the formation of fat and inflammation in the liver | [11] |
| Cheddar | *Lactobacillus acidophilus* and distinct mesophilic starter cultures | To evaluate the antithrombotic efficacy of buffalo milk probiotic cheese | The water-soluble extract of probiotic cheddar cheese showed greater antithrombotic activity compared to the control cheese, and the activity increased with the ripening period. | [91] |
| Fior di Latte-type | *Lactobacillus rhamnosus* GG and *Lactobacillus acidophilus* LA5 | To study the impact of adding *Lactobacillus* probiotics in cheese (Fior di Latte-type) on evaluating their immunomodulatory capacity using an in vivo murine model. | Probiotic cheeses (with individual or combined strains) were able to modulate the immune system of mice, reducing the secretion of pro-inflammatory cytokines in the intestine, and increasing the secretion of secretory IgA (S-IgA) | [92] |

In a study with the probiotic strain *Lacticaseibacillus casei*-01 incorporated into Prato cheese (a Brazilian cured cheese), Vasconcelos et al. [6] observed that repeated ingestion of probiotic cheese by mice exposed to cigarette smoke was able to reduce oxidative stress in the lungs, intestine, and liver, and alleviated inflammation in the lungs, and these effects may be related to the ability of probiotic strains to reduce reactive oxygen species probably due to the production of antioxidant enzymes.

Mudgil et al. [66] incorporated probiotic cells (*Pediococcus pentosaceus*) microencapsulated in camel milk protein and wheat starch into Chami cheese (traditional Emirati soft cheese) and evaluated the storage stability and survival of probiotics in the gastrointestinal

passage. In the study, the authors observed that cheeses fortified with probiotic bacteria encapsulated in camel milk protein exhibited greater retention of bioactive properties during storage, in terms of inhibition of α-glucosidase, and Dipeptidyl peptidase IV (DPP-IV), which are enzymatic markers involved in diabetes, thus indicating potential in controlling this comorbidity.

Carvalho et al. [93] developed two experimental Swiss-type cheeses: an Emmental cheese prepared with two LABs (*Lactobacillus delbrueckii* subsp *lactis* CNRZ327 and *Streptococcus thermophilus* LMD-9) combined with a probiotic strain (*Propionibacterium freudenreichii* CIRM-BIA129); and one a single strain cheese (*P. freudenreichii* CIRM-BIA129). The authors evaluated the protective capacity of the two cheeses in a murine colitis (an inflammatory bowel disease) model induced by dextran sodium sulphate (DSS) which acts as a direct chemical toxin on the colon epithelium, resulting in epithelial cell damage in mice. Consumption of single-strain cheese restored some of the impaired metabolic functions of the microbiome, while Emmental cheese promoted an increase in the microbiota's ability to produce metabolites with neuromodulation properties and promoted an increase in *Ligilactobacillus murinus* (promising bacteria in the treatment of intestinal inflammatory disorders and relief of DSS-induced colitis) compared to single-strain experimental cheese (*P. freudenreichii* CIRM-BIA129).

Probiotic LAB can also secrete active compounds that are byproducts of their metabolism, such as essential vitamins, enzymes, bacteriocins, antioxidants, bioactive peptides, and short-chain fatty acids, which have a direct impact on the host's health [84,94,95].

The beneficial effects of the generation of bioactive peptides through the proteolytic activity of probiotic strains have attracted attention and stimulated the use of probiotics to induce casein hydrolysis and the release of these peptides during cheese manufacturing and ripening. Baptista et al. [61] used the strain *Lactobacillus helveticus* LH-B02 in Prato cheese, which resulted in favoring the inhibitory activity of the angiotensin-converting enzyme (ACE) during cheese maturation, indicating that bioactive peptides can play an important role in reducing blood pressure. In addition to the mentioned inhibitory activity, *L. helveticus* is known for its antioxidant, antimicrobial, and immunomodulatory effects, which mainly result from its ability to produce functional peptides [96].

Cordeiro et al. [5] developed a Minas Frescal cheese with the probiotic strain of *Lactococcus lactis* NCDO 2118 and evaluated the therapeutic effect of consuming this cheese in rats, in combating colitis. In vivo analyses showed that this type of probiotic cheese was able to alleviate the severity of colitis induced by dextran sodium sulfate (DSS) in the intestinal barrier of rats. Minas Frescal probiotic cheese was also able to prevent the degeneration of goblet cells and reduce the infiltration of inflammatory cells in the colon mucosa. The experimental probiotic cheese investigated in this work was also capable of producing high levels of bioactive peptides with antihypertensive, antioxidant, and antidiabetic activities. Mushtaq et al. [88] also reported that the increase in immunomodulatory, antidiabetic, and antimicrobial activity in water-soluble extract from Kradi/Kalari cheese with the addition of probiotics might be related to the bioactive peptides generated.

The publications discussed in this work prove that cheese is an excellent matrix for adding probiotic microorganisms, thus leading it to a potential functional food. Increasing the production and study of probiotic cheeses and their benefits will provide consumers with a greater diversity of safe foods that benefit their health.

## 8. Conclusions

In this review, an attempt is made to summarize current knowledge about the innovations and benefits associated with the production and consumption of probiotic cheeses. It was possible to notice the growing number of studies focused on the use of these cheeses and their relationship with their health benefits. However, despite technological advances, the survival of probiotic microorganisms during the production, storage, and transportation process to the target location is still a challenge. Thus, the development and adoption of

technologies with the aim of increasing the viability of probiotic bacteria are necessary and increasingly explored.

Another point about probiotic research is the limited knowledge about the complex mechanisms of health benefits resulting from the actions of probiotics. Most in vivo studies reported in the literature have been proposed and explored in an animal model, and considering that animal models cannot completely recapitulate human biology, it is important and necessary to carry out careful and safe clinical trials in humans, in order to determine viability and better understand the beneficial effect on the host.

**Author Contributions:** Conceptualization, H.C.S.A.; resources, N.N.; writing—original draft preparation, H.C.S.A., M.S.d.J., M.T.S.L.N., R.D.D.S. and N.N.; writing—review and editing, N.N.; visualization, N.N.; supervision, N.N.; project administration, N.N.; funding acquisition, N.N. All authors have read and agreed to the published version of the manuscript.

**Funding:** This research was funded by CNPq (Conselho Nacional de Desenvolvimento Científico e Tecnológico), Brazil, vide research project Instituto Nacional de Ciência e Tecnologia de Frutos Tropicais (Project 465335/2014-4) in developing this work, for their fellowships.

**Institutional Review Board Statement:** Not applicable.

**Informed Consent Statement:** Not applicable.

**Data Availability Statement:** Data available on request.

**Conflicts of Interest:** The authors declare no conflict of interest.

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
