# Peer review of "Functional Cheeses: Updates on Probiotic Preservation Methods"

_fermentation, doi:10.3390/fermentation10010008_

Round 1
Reviewer 1 Report (New Reviewer)
Comments and Suggestions for Authors
Comments to authors
Journal: Fermentation-2739903
Title: ‘Innovations in cheese as functional food’
The current article highlights the rising popularity of functional foods, particularly probiotics, emphasizing their health benefits. It explores the significance of probiotic-enriched cheeses, discussing their potential and the need for advancements in technology to optimize probiotic viability, making it a promising area for further research. The article is looks like a collection of date but nothing novel. Still, I have fewer questions as mentioned below.
comments:
1. The current title of the article is should be change. I suggest to make it more captivating and indicative of the article's innovative insights. Perhaps consider a title that reflects the pioneering advancements in utilizing cheese for its health benefits while intriguingly highlighting its evolution in the functional food landscape.
2. More detail required in the introduction.
3. The article is focused on probiotics therefore add the word probiotics in the title.
4. What is novelty of this article? I suggest you to emphasize your article with unique angle because it looks like a fresh take on existing data, a new perspective, or the latest advancements integrated into the context of functional cheese. Already there many articles have been published on the same contents.
5. Does the authors' assertion of cheese as a functional food, particularly its utilization of probiotics, convincingly support its potential health benefits, transforming its traditional image into a functional dietary component?
6. The language should be improved and authors should build a connection among the sentence.
7. Including relevant, illustrative figures can significantly enhance the article's quality and captivate readers. So, I suggest the authors to add some figures in the article.
8. Add few sub-headings of functional foods used in the market under the heading of functional foods.
Comments on the Quality of English Language
Need to be revised by native speaker
Author Response
We appreciate the comments made by the Reviewer and thank you for your valuable suggestions. We wish to inform that all these comments have been duly considered in the preparation of new manuscript. Thus, revision of the full paper has been done thoroughly. The paper presentation has been substantially modified and English language in the text has been corrected by a native-English speaking person.

Reviewer 2 Report (New Reviewer)
Comments and Suggestions for Authors
Respected authors!
After reading the paper, it is evident that a lot of effort has been invested in reviewing the literature and writing this work. Nonetheless, I believe that the title of the paper does not correspond well to its content. The paper exclusively addresses the addition of probiotic bacteria to cheese in various forms. Other recent popular innovations have not been explored in the paper. For instance, the incorporation of plant extracts into cheese to enhance antioxidant activity and inhibit undesirable microflora, the use of biofilms and edible films in cheese coating, also with plant extracts, and many other innovations have not been covered.
Furthermore, the paper contains too much general knowledge that is more suitable for textbooks rather than review articles. There is an excessive description of probiotic bacteria and the methods of their production. My recommendation is to either change the title to better match the content of the paper or to include additional innovations in cheese production.
Best wishes!
Author Response
We appreciate the comments made by the Reviewer and thank you for your valuable suggestions. We wish to inform that all these comments have been duly considered in the preparation of new manuscript. Thus, revision of the full paper has been done thoroughly. The paper presentation has been substantially modified and English language in the text has been corrected by a native-English speaking person.

Round 2
Reviewer 1 Report (New Reviewer)
Comments and Suggestions for Authors
The Authors have improved the articles as per suggestions, But the title of the article is still no attractive. I suggest the authors to improve the title according to the manuscript details.
Comments on the Quality of English LanguageEnglish is fine.
Author Response
We appreciate the comments made by you and thank you for their valuable suggestions.

Reviewer 2 Report (New Reviewer)
Comments and Suggestions for Authors
The paper can be published now!
Author Response
We appreciate the comments made by you and thank you for their valuable suggestions.

This manuscript is a resubmission of an earlier submission. The following is a list of the peer review reports and author responses from that submission.
Round 1
Reviewer 1 Report
Comments and Suggestions for Authors
Dear Authors,
Thank you for submitting the review articles! The articles appears to be well curated but does not appears an intensive review of the processes on technological advances for probiotics in cheese. There were several instances where the key information was missing. There was no pictorial representation of any process to make it easier for the readers to comprehend well. Please see below the comments:
1. The authors mentioned that there is a requirement for a certain count of viable cells to be included. However, the authors did not mention counts for any cheese and probiotic culture included.
2. The authors did not mention about aging. It is one of the important aspect in the certain cheeses. Usually, aging sometimes involve the use of microbes. So, it was unclear which cheese underwent aging and which not. And it the microbe involved in the aging was different than probiotic culture.
3. The author did not mention about the timing of addition of the probiotic culture to the cheese making process.
4. Table 1 label does not resonate well with the information provided in the table.
5. There is no description of the parameters that need to be controlled when adding probiotic cultures to cheese.
6. How the cheese is curated after adding probiotic cultures? Are there any processing steps required after addition of probiotic cultures.
7. There is no mention of the stability of the probiotic cultures in the cheese.
8. In section 4.1 there is generalization of term probiotic as the microorganisms that are beneficial. It sounds like it includes the beneficial microbes present in human body also, whereas the probiotic involved only the microbes that are administered. The authors also mentions this in the next line, but is contradictory here.
9. No process variables are mentioned in different technology for preserving probiotics (temperature, pH etc.). High temperature can be 37 or 60 C depending upon the microbe used.
10. The technology section covers a majority and is one of the highlights. However, the authors did not include a table comparing the pros and cons with different methods?
Thanks
Author Response
Thank you very much for taking the time to review this manuscript. Please find the detailed responses below and the corresponding revisions/corrections highlighted/in track changes in the re-submitted files.

Reviewer 2 Report
Comments and Suggestions for Authors
Summary:
In the review by Araujo et al, the authors attempted to summarize the probiotics strains presented in cheese, the preservation methods for the probiotics in cheese, and the beneficial effects of probiotics in cheese. The manuscript covered adequate amount of recent publications in the field. It was concluded that the development and adoption of new technologies could increase the viability of probiotic bacteria in cheese.
The authors did a great job in terms of extracting relevant information from recent publications. However, although I believe this is a very interesting topic and the review is valuable to the scientific community, some of the information in the manuscript were not presented in a well-structured manner. The connections between the three main sections of the of the paper (cheese, health benefits of probiotics, and preservation methods) are rather loss. The review is entitled “innovation in cheese” but it is not cheesy enough, I could have mistaken it as a review for LAB/probiotics. There is no denying that many of the relevant information could be found in the tables, which were well-designed, comprehensive, and easy to interpret. If some of the facts mentioned in the tables could be incorporated into the text, the different parts of the paper would drawn coherent.
In conclusion, while this review covered some recent research progresses in the field, the way these information was summarized makes it slightly less convincing. It could be reconsidered after revision.
Major issues:
· Title of the manuscript and the headings of each section could be rewritten to make the structure of the manuscript clearer. For example, both cheese and functional foods are very broad concept, these section titles could be more specific. The probiotics section could be breakdown further, at lease there could be a section 4.0 in parallel to section 4.1.
· Indigenous strains from milk were mentioned several times in the review, but if pasteurized milk is used in cheese production, the impacts of indigenous strains on final properties should be justified.
· Microencapsulation: the mechanism for encapsulation was not well-explained. In the text, extrusion, spray-drying, and lyophilization were discussed as preservation methods, but how encapsulation happened during the above processes? Maybe a figure could be used to demonstrate the encapsulation mechanism for all the methods mentioned.
· Still for the microencapsulation part, the word cheese and dairy only appeared few time. Some details regarding how cheese fits into each of the preservation method could be added.
· Is there any research focusing on in vitro digestion of the encapsulated cheese?
· Line 168-182 and line 202-210 are redundant.
· When referring to results from previous work, a brief description of is needed to emphasise the importance of the cited work, and why they are cited here (e.g. line 436-line 448 is a little bit hard to follow, but line 453-line462 is clear).
Questions:
1. Apart from bacteria presented in starter culture primarily for fermentation purposes, is there any strain of probiotics added into cheese mainly as probiotics?
2. What is the relationship between probiotic strains and gut microbiota?
Some specific issues:
Writing: Use comma for numbers with 4 or more digits.
Line 14 - line 19: This sentence is too long.
Line 151 - 152: Butter is not commonly accepted as functional food.
Line 388: letter A is bolded.
Line 411: “high density” is not clear.
Line 482-483: not clear.
Comments on the Quality of English Language
There are some awkward expressions. They are ok to understand, but not intuitively understandable. Some careful revision of English would help.
Author Response
Detailed response to Reviewers comments
Ms. Ref. No.: Fermentation-2668038
Title: Innovations in cheese as functional food.
General Comments:
We appreciate the comments made by the Reviewers and thank them all for their valuable suggestions. We wish to inform that all these comments have been duly considered in the preparation of new manuscript. Thus, revision of the full paper has been done thoroughly. The paper presentation has been substantially modified and English language in the text has been corrected by a native-English speaking person.
Reply on Reviewers comments:
(Reviewer´s comments are reproduced below in Normal format while our reply & rebuttal is presented in blue color text. Please note that the comments of Reviewers refer to the pages and line numbers of the old manuscript while our reply refers to the page numbers of the new corrected manuscript.
Reviewer #2:
Title of the manuscript and the headings of each section could be rewritten to make the structure of the manuscript clearer. For example, both cheese and functional foods are very broad concept, these section titles could be more specific. The probiotics section could be breakdown further, at lease there could be a section 4.0 in parallel to section 4.1.
- We appreciate your observation and the suggested changes were made. To make the text more cohesive and fluid, we modified some subtitles and changed their order, as can be seen below:
Line 27: “1. Introduction”
Line 58: 2. Functional Foods
Line 99: 3. Probiotics
Line 126: 3.1. Probiotic Strains
Line 139: 4. Probiotic Cheese
Line 249: 5. Technology for preserving probiotics in cheese
Line 275: 5.1. Extrusion
Line 287: 5.2. Spray-drying
Line 304: 5.3. Lyophilization/ Freeze drying
Line 319: 5.4. Emulsion
Line 334: 5.5. Coacervation
Line 371: 6. Current applications of probiotics in cheese as health promoters
Line 459: 7. Conclusion
Indigenous strains from milk were mentioned several times in the review, but if pasteurized milk is used in cheese production, the impacts of indigenous strains on final properties should be justified.
- We appreciate your observation.
Studies that report the use of indigenous strains isolated from raw milk incorporate these strains in the processing of cheeses after milk pasteurization, such as stated in the following work:
de Moraes et al. (2018) used the probiotic strain Lb. mucosae CNPC007, isolated from goat's milk in the production of Coalho goat cheese. Firstly, this strain was freeze-dried and was incorporated, together with the starter culture, after pasteurization and subsequent cooling of the milk (35°C).
de Moraes, G. M. D.; dos Santos, K. M. O.; de Barcelos, S. C.; Lopes, S. A.; do Egito, A. S. (2018). Potentially probiotic goat cheese produced with autochthonous adjunct culture of Lactobacillus mucosae: Microbiological, physicochemical and sensory attributes. Lwt, 2018, 94, 57-63. https://doi.org/10.1016/j.lwt.2018.04.028
Microencapsulation: the mechanism for encapsulation was not well-explained. In the text, extrusion, spray-drying, and lyophilization were discussed as preservation methods, but how encapsulation happened during the above processes? Maybe a figure could be used to demonstrate the encapsulation mechanism for all the methods mentioned.
- We appreciate your observation. The following excerpts from the text have been rewritten to improve the explanation of the principles of each method covered:
Line 275- 286:
“Extrusion
Extrusion is a physical method that is based on mixing probiotic microorganisms with a hydrocolloid solution to form a suspension, which is then extruded through a nozzle at high pressure using a syringe or appropriate droplet generating devices. The droplets produced are placed in a gelling bath and from this direct contact, the cells become trapped in a three-dimensional grid that occurs from the ionic cross-linking of the polymer, thus forming gel spheres that can be incorporated into food matrices [46,55].
Alginates and other natural polysaccharides are often used as hydrocolloid materials because they do not cause damage to probiotic cells, thus resulting in high viability in the food product and prolonged survival of encapsulated cells under storage and simulated gastric and enteric environments. As for the gelling (or hardening) solution, it consists of divalent cations, such as magnesium or, more commonly, calcium [46,55,56].”
Line 287- 303:
“Spray drying
Spray drying is one of the most commonly used encapsulation techniques for food ingredients. The encapsulation of probiotics using this method is based on the emulsion or dispersion of the strains in an aqueous solution with the encapsulating agent, followed by spraying the mixture through a nozzle and evaporating the water from the contact of the droplets with air or hot gas which results in a dry particulate powder with low water activity [46,55].
Studies on encapsulation of microorganisms and probiotic products by spray drying have increased, and different authors have suggested that this process has improved the stability and viability of probiotics, and is more efficient in the long-term preservation of lactic acid and other probiotic cultures encapsulated with different materials support [46]. However, the effectiveness of encapsulation depends on different factors, such as: probiotic characteristics (probiotic strain and its growth phase); dryer characteristics and spray drying conditions (flow rate, chamber humidity, chamber inlet and outlet temperatures); addition of encapsulating material and/or protectants (trehalose, non-fat milk solids (MSNF), adonitol, acacia, granular starch, prebiotics, soluble fiber, maltodextrin, gum Arabic, etc.) and solution viscosity, and so on) [46,55,57].”
Line 304-318:
“Lyophilization/ Freeze drying
Freeze drying is one of the most commonly used techniques for dehydrating probiotic cultures and other thermally sensitive components, and is based on the freezing and subsequent sublimation of the frozen ice present in the molecules to the gaseous state, under vacuum and low temperatures [55,58]. Freeze-dried probiotic microorganisms are very stable and have a high survival rate (higher compared to other methods such as spray drying), and have been extensively used in many dairy products [46].
Despite the effectiveness of the method, freeze-drying affects the state of cell membrane lipids and the structure of sensitive cellular proteins, thus having a negative effect on the survival of the probiotic [55]. Studies are being carried out to improve the efficiency of lyophilization of these strains, and one of the alternatives is the combination of lyophilization with suitable encapsulation agents and/or cryoprotectants [55,56]. One of these examples is the first work carried out that reported the encapsulation of Lactobacillus F19 and Bifidobacterium Bb12 in enzymatically gelled sodium caseinate microcapsules, followed by freeze-drying [59].”
Line 319 -333:
“Emulsion
Emulsification is a technique that can be used in food processing to encapsulate probiotic microorganisms. The principle of this technique is the dispersion of a liquid (a small volume of a water-based polymeric suspension containing probiotic cells) in another immiscible liquid (a large amount of oil/organic phase) followed by mechanical homogenization in the presence of an emulsifier to form a water-in-oil emulsion. Gel spheres are formed in the oil phase as the water-soluble biopolymer becomes insolubilized [55].
The emulsion system methods that can be used to encapsulate probiotics can be divided into three categories: a) emulsification and ionic gelation; b) emulsification and enzymatic gelation; c) emulsification and interfacial polymerization. Ding et al. [60] generated W/O/W emulsions with the internal aqueous phase loaded with probiotic cells and various protective compounds. The oil phase consisted of soybean oil and polyglycerol polyricinoleate (PGPR), a common food-safe molecular surfactant, stabilizing the W1/O and O/W2 interfaces [60,61].”
Line 334 -351:
“Coacervation
Coacervation is the highly promising and widely used encapsulation technology for probiotics and consists of three steps: the formation of immiscible phases, the deposition of the coating, and the solidification of the coating. Probiotic cells are added to a hydrocolloid and the main polymers used as encapsulating agents are proteins and various polysaccharides [55].
This technology can be conducted by two methods: simple and complex coacervation. The first consists of evaporating the solvent that surrounds the molecules of a colloid, through the addition of another non-electrolytic solvent in which the colloid is insoluble. Complex coacervation consists of the combination of two hydrocolloid solutions with opposite charges, causing interaction and precipitation of complex polymers [46,54]. Some parameters must be considered for the method to be effective in encapsulation, such as the total percentage of solids, the dispersion conditions (ionic strength, pH, temperature, etc.), the characteristics of the encapsulating materials and core ingredients (weight molecular, conformation, charge density and composition) [55].”
The following references were added in the manuscript:
- Frakolaki, G.; Giannou, V.; Kekos, D.; Tzia, C. A review of the microencapsulation techniques for the incorporation of probiotic bacteria in functional foods. Crit. Rev. Food Sci. Nutr. 2021, 61, 1515-1536, https://doi.org/10.1080/10408398.2020.1761773.
- Wu, Y.; Jha, R.; Li, A.; Liu, H.; Zhang, Z.; Zhang, C.; Zhai, Q.; Zhang, J. Probiotics (Lactobacillus plantarum HNU082) Supplementation Relieves Ulcerative Colitis by Affecting Intestinal Barrier Functions, Immunity-Related Gene Expression, Gut Microbiota, and Metabolic Pathways in Mice. Microbiol. Spectr. 2022, 10, e01651-01622, https://doi.org/10.1128/spectrum.01651-22.
Still for the microencapsulation part, the word cheese and dairy only appeared few time. Some details regarding how cheese fits into each of the preservation method could be added.
- We appreciate your observation. Table 4 presents a summary of different publications related to the preservation methods mentioned in the text that were applied to different types of cheese, as well as the main observations regarding the effectiveness of the method.
Is there any research focusing on in vitro digestion of the encapsulated cheese?
- References found in the literature and cited in the manuscript ([51], [13], [69], [31] [66]) are main objective to evaluate the viability of encapsulated probiotic strains added to cheeses, through in vitro
El Sayed [51] evaluated the viability of probiotics microencapsulated with rice flour in simulated gastrointestinal conditions, within UF-Kariesh cheese and reported that strains of Lactobacilli and Bifidobacterium microencapsulated with sodium alginate and rice flour showed a survival rate in simulated stomach conditions of 72.91, 68.43, 61.27 and 59.23% for microcapsule strains with 5%, 3%, 1% rice flour with alginate, respectively.
de Andrade et al. [13] evaluated the effect of microencapsulation of L. plantarum CCMA 0359 using the spray drying technique on the survival of the probiotic culture during refrigerated storage and in vitro digestion when incorporated into cream cheese. They reported that in general, the viability of L. plantarum CCMA 0359 increased slightly by 0.33 log CFU/g, for microencapsulated cells when compared to the viability of free cells, during simulated gastrointestinal transit.
Sharifi et al. [69] investigated the viability of the probiotic bacterium Lactobacillus plantarum ATCC 8014, during in vitro digestion, microencapsulated and co-encapsulated with phytosterols added in ultrafiltered Iranian white cheese. They concluded that the population of microencapsulated and coencapsulated probiotics of L. plantarum with phytosterols at the end of the storage period reached 7.95 and 8.14 Log CFU/g, respectively.
Motalebi Moghanjougi et al. [31] studied the bio-preservation of white brined cheese (Feta) coated with bacterial cellulose film (BCF) containing Lactobacillus acidophilus and Bifidobacterium animalis microencapsulated with sodium alginate (AS) and pectin and demonstrated that the viability of these probiotic microorganisms increased drastically when microencapsulated.
Estilarte et al. [66] studied the preservation of probiotic strains of Enterococcus durans by freeze-drying added to a commercial un-ripened cream cheese. They reported that freeze-dried E. durans LM01C01 maintained the evaluated probiotic and technological properties, with resistance to gastrointestinal conditions (acid and bile) and tolerance to salts (NaCl and potassium sorbate), respectively.
Line 168-182 and line 202-210 are redundant.
- We appreciate your observation. The redundant sections were removed from the text.
When referring to results from previous work, a brief description of is needed to emphasize the importance of the cited work, and why they are cited here (e.g. line 436-line 448 is a little bit hard to follow, but line 453- line462 is clear).
- We appreciate your observation. The following excerpts of the text have been rewritten and improved:
Lines 409- 416: “Mudgil et al. [64] incorporated probiotic cells (Pediococcus pentosaceus) microencapsulated in camel milk protein and wheat starch into Chami cheese (traditional Emirati soft cheese), and evaluated the storage stability and survival of probiotics in the gastrointestinal passage. In the study, the authors observed that cheeses fortified with probiotic bacteria encapsulated in camel milk protein exhibited greater retention of bioactive properties during storage, in terms of inhibition of α-glucosidase and Dipeptidyl peptidase IV (DPP-IV), which are enzymatic markers involved in diabetes, thus indicating potential in controlling this comorbidity.”
Lines 417-429: “Carvalho et al. [85] developed two experimental Swiss-type cheeses: an Emmental cheese prepared with two LABs (Lactobacillus delbrueckii subsp lactis CNRZ327 and Streptococcus thermophilus LMD-9) combined with a probiotic strain (Propionibacterium freudenreichii CIRM-BIA129); and one a single strain cheese (P. freudenreichii CIRM-BIA129). The authors evaluated the protective capacity of the two cheeses in a murine colitis (an inflammatory bowel disease) model induced by dextran sodium sulphate (DSS) which acts as a direct chemical toxin on the colon epithelium, resulting in epithelial cell damage in mice. Consumption of single-strain cheese restored some of the impaired metabolic functions of the microbiome, while Emmental cheese promoted an increase in the microbiota's ability to produce metabolites with neuromodulation properties and promoted an increase in Ligilactobacillus murinus (promising bacteria in the treatment of intestinal inflammatory disorders and relief of DSS-induced colitis) compared to single-strain experimental cheese (P. freudenreichii CIRM-BIA129).”
Apart from bacteria presented in starter culture primarily for fermentation purposes, is there any strain of probiotics added into cheese mainly as probiotics?
- Studies covered in the text use strains that act only as probiotics and are not part of starter cultures, such as:
Cordeiro et al. [5] investigated the therapeutic effects of fresh Minas cheese added with the probiotic microorganism Lactococcus lactis NCDO 2118. According to Oliveira et al. (2014), this microorganism is a non-dairy strain, fermenting xylose (a common characteristic of plant-associated strains) and producing gamma-aminobutyric acid isolated from frozen peas. This microorganism is only a probiotic with anti-inflammatory and immunomodulatory activity in the treatment of diseases, mainly inflammatory bowel diseases.
Hurtado-Romero et al. [52] developed and characterized two blueberry-based ingredients, a concentrated syrup and a freeze-dried pomace powder, and incorporated them into a synbiotic Petit Suisse cheese containing probiotics and inulin. Lactobacillus acidophilus LA5 and Bifidobacterium animalis subsp. lactis BB12 were the probiotic strains used in this study. These strains originate from the Christian Hansen company's dairy culture collection and are marketed as probiotic strains.
Table 2 discussed in the manuscript and focused on the probiotic microorganisms that are also used as starter cultures. As reported in the publications above, there are strains that are added to cheeses only as probiotics and are not part of the starter culture.
Oliveira, L.C.; Saraiva, T.D.; Soares, S.C.; Ramos, R.T.; Sá, P.H.; Carneiro, A.R.; Miranda, F.; Freire, M.; Renan, W.; Júnior, A.F.; Santos, A.R.; Pinto, A.C.; Souza, B.M.; Castro, C.P.; Diniz, C.A.; Rocha, C.S.; Mariano, D.C.; de Aguiar, E.L.; Folador, E.L.; Barbosa, E.G.; Aburjaile, F.F.; Gonçalves, L.A.; Guimarães, L.C.; Azevedo, M.; Agresti, P.C.; Silva, R.F.; Tiwari, S.; Almeida, S.S; Hassan, S.S.; Pereira, V.B.; Abreu, V.A.; Pereira, U.P.; Dorella, F.A.; Carvalho, A.F.; Pereira, F.L.; Leal, C.A.; Figueiredo, H.C.; Silva, A.; Miyoshi, A.; Azevedo, V. Genome sequence of Lactococcus lactis subsp. lactis NCDO 2118, a GABA-producing strain. Genome Announc., 2014, 2, 10-1128, doi: 10.1128/genomeA.00980-14.
What is the relationship between probiotic strains and gut microbiota?
- As reported in the text:
Lines 100-108: “The human gastrointestinal tract is the habitat of several bacteria, where a symbiotic relationship is maintained with the rest of the body. The intestinal microbiota is a complex ecosystem composed of microorganisms linked to various nutritional, metabolic, endocrine, psychological and immunological mechanisms. These microorganisms have the functions of maintaining the intestinal barrier, nutrient metabolism, synthesis of bioactive compounds and vitamins, fermentation of non-digestible carbohydrates and mainly immunomodulation. The so-called “barrier effect” that prevents pathogens from affecting the gastrointestinal tract can be strengthened by taking dietary supplements such as probiotics [24,25].”
Line 14 - 19: This sentence is too long
- We appreciate your observation. We modified the text to the following:
Lines 14-19: “Probiotic dairy products, especially cheese, are the most appreciated and have been produced to possess the properties that favor the viability of these microorganisms after passing through the gastrointestinal tract. They provide benefits such antibacterial activity; prevention of cancer and cardiovascular diseases; anti-obesity effect; anti-diabetic effect; improve the immune system; promote balance in the intestinal microbiota; reduction of symptoms of lactose intolerance; allergic diseases, among others.”
Line 151 - 152: Butter is not commonly accepted as functional food.
- We appreciate your observation. We deleted the requested word in the lines 88-91.
Line 388: letter A is bolded.
- We appreciate the comment and have now corrected in the line 360.
Line 411: “high density” is not clear.
- We thank you for your comment. The expression “high density” refers to the dense structure of the cheese and this characteristic promotes the protection of probiotic microorganisms during their passage through the gastrointestinal tract. Therefore, the following excerpt in the text was rewritten:
Lines 383 – 388: “Cheese is one of the best and most favorable food matrices for the incorporation of probiotics. In addition to being one of the most consumed dairy products in the world, it has a high buffering capacity for milk proteins, a high fat content, lactose as a fermentable sugar and the dense structure of this dairy matrix, which encourages the survival of probiotic strains during storage and through passage in the gastrointestinal tract [73,75].”
Line 482-483: not clear.
- We thank you for your comment. With this sentence described in lines 462-465, we wanted to show that although several studies are being developed on the incorporation of probiotics in cheeses and consequently the study of their viability, the survival of these microorganisms is still an obstacle, due to storage and transport conditions that probiotic cheese can be subjected. Thus, the development of technologies that protect these microorganisms becomes increasingly important and
Round 2
Reviewer 2 Report
Comments and Suggestions for Authors
Comments:
The quality of the manuscript and English fluency improved significantly after revision. The questions posted in the first review were answered in a satisfactory way. It can be published after some minor revision.
One suggestion:
Maybe the authors can briefly describe at which stage(s) microencapsulation/encapsulated probiotics can be applied in cheese manufacturing.
Minor/editorial issues (line number in clean copy):
Line 113-114, 210-211: Use comma for numbers with 4 or more digits.
Line 167-187: The usage of oxford comma is not consistent throughout the manuscript.
Line 209-213: Tons and metric tons. If they are the same, keep it constant; if not, convert one to the other. .
Table 3: Need space or a line between different techniques to separate them. “Disadvantages” and “Benefits” could be centered. Anyway, the table is a little hard to read due to its format.
Table 4: For ionic gelation, if possible, it’s better to describe the process conditions with the same parameter. Also, the authors could a little more specific on “biopolymeric solution”.
Author Response

(The authors gave the same response as above.)
